# Development of Self-Sensing Asphalt Pavements: Review and Perspectives

**DOI:** 10.3390/s24030792

**Published:** 2024-01-25

**Authors:** Federico Gulisano, David Jimenez-Bermejo, Sandra Castano-Solís, Luis Alberto Sánchez Diez, Juan Gallego

**Affiliations:** 1Departamento de Ingeniería del Transporte, Territorio y Urbanismo, Universidad Politécnica de Madrid, C/Profesor Aranguren 3, 28040 Madrid, Spain; juan.gallego@upm.es; 2Information Processing and Telecommunication Center (IPTC-GATV), Universidad Politécnica de Madrid, 28040 Madrid, Spain; david.jimenezb@upm.es; 3Escuela Técnica Superior de Ingeniería y Diseño Industrial (ETSIDI), Universidad Politécnica de Madrid, Ronda de Valencia 3, 28012 Madrid, Spain; sp.castano@upm.es; 4Departamento de Ingeniería Civil, Hidráulica, Energía y Medio Ambiente, Universidad Politécnica de Madrid, C/Profesor Aranguren 3, 28040 Madrid, Spain; luisalberto.sanchez@upm.es

**Keywords:** asphalt pavements, self-sensing, structural health monitoring, digitalization, transportation

## Abstract

The digitalization of the road transport sector necessitates the exploration of new sensing technologies that are cost-effective, high-performing, and durable. Traditional sensing systems suffer from limitations, including incompatibility with asphalt mixtures and low durability. To address these challenges, the development of self-sensing asphalt pavements has emerged as a promising solution. These pavements are composed of stimuli-responsive materials capable of exhibiting changes in their electrical properties in response to external stimuli such as strain, damage, temperature, and humidity. Self-sensing asphalt pavements have numerous applications, including in relation to structural health monitoring (SHM), traffic monitoring, Digital Twins (DT), and Vehicle-to-Infrastructure Communication (V2I) tools. This paper serves as a foundation for the advancement of self-sensing asphalt pavements by providing a comprehensive review of the underlying principles, the composition of asphalt-based self-sensing materials, laboratory assessment techniques, and the full-scale implementation of this innovative technology.

## 1. Introduction

Road infrastructure is one of the most important and valuable assets of a country due to its role in facilitating communication, the transportation of goods and people, and driving economic, cultural, and social development [1]. The responsibility of road administrations lies in the maintenance, operation, improvement, and preservation of this asset while effectively managing limited financial and human resources [2,3]. Special attention should be paid to the preservation of road asphalt pavements, as they represent a key element for ensuring the safe and efficient circulation of road users. Moreover, due to pavements’ limited lifespans and exposure to harsh conditions such as traffic loads, temperature variation, and moisture, significant efforts are required for pavement maintenance and rehabilitation.

In this regard, the implementation of appropriate pavement management systems (PMSs) is essential. PMSs are a set of tools and methods that aid decision-makers in determining optimal strategies for providing, evaluating, and maintaining pavements in a serviceable condition over time. PMSs are necessary for the transition from a reactive to a proactive (e.g., preventive or predictive) maintenance approach. However, the correct implementation of PMSs requires the collection of a large quantity of data from different sources regarding pavement usage (i.e., traffic loads), structural and functional conditions, environmental variables, etc. The massive quantities of data collected are therefore fed into a specific software product that uses statistical analysis for the development of reliable pavement deterioration models and the selection of the optimum maintenance strategy.

In this context, carrying out systematic monitoring campaigns is of paramount importance for collecting useful information and insights for analytics [4]. Systematic auscultation campaigns typically consist of visual inspections and the employment of specially equipped vehicles for collecting data about the bearing capacity and surface properties of pavement [5]. Nevertheless, traditional monitoring operations require significant amounts of manpower and time and the use of costly equipment; in addition, they are not able to detect micro-damage and its development [6]. For this reason, many efforts have been made in the last decades to implement more advanced and effective sensing technologies and non-destructive techniques (NDTs) able to collect information about traffic loads and the health condition of pavement continuously and in real-time [7].

The use of sensors embedded in asphalt pavement enables the acquisition of data about the stress, strain, and deflection of pavement and these factors’ evolution over time, constituting information necessary for the development of PMS [8]. The real-time approach enables the implementation of more-informed decision support systems and applications for Digital Twins [9]. Traditional embedded sensors include strain gauges, linear variable differential transformers (LVDTs), accelerometers, load cells, geophones, and thermocouples [5,10]. More recently, the use of Wireless sensor networks (WSNs) [6,11] and fiber optic sensor technologies [12,13,14] showed promising results. Several full-scale projects have demonstrated the feasibility of embedding intrusive sensors inside pavement for monitoring operations. The NCAT test track [15], the Virginia smart road [16], MnRoad [17], and the PEGASE instrumentation platform [18] are only some examples of infrastructures instrumented with embedded sensors and validated at full-scale. Although the aforementioned intrusive sensors have been shown to provide reliable pavement-monitoring data, their main drawbacks concern the risk of premature damage due to the harsh climatic conditions to which these sensors are subjected during pavement’s lifespan. First, the embedded sensors must withstand the high temperature and stress experienced during the construction process [19]. Then, during their service lives, traffic load and climatic conditions also affect the survival rates of the sensors [20]. In addition, a lack of compatibility between the sensor and the host material (i.e., asphalt mixture) can also lead to localized mechanical discontinuities and the premature failure of road pavement [21].

For this reason, the search for new cost-effective, high-performing, and durable sensing pavement technology is crucial. To overcome the limitations of embedded sensors, the concept of self-sensing road pavement, also known as self-monitoring or self-diagnosing pavement, has been developed. Self-sensing refers to the ability of a structural material to sense itself without the need for embedded sensors [22]. These materials are designed by incorporating a certain quantity of conductive additives in such a way as to form an extensive conductive network inside a structural material and endow it with self-sensing functions while maintaining or even enhancing the mechanical and durability performance of the pavement [23]. The application of external stimuli (stress/strain), the presence of damage, or climatic factors (temperature or moisture) disturb the conductive network and lead to a change in the electrical properties of the material. Therefore, by measuring changes in electrical properties, the external stimulus can be detected. The intrinsic sensing properties of these smart materials make their use preferable over traditional sensors [24,25]. Self-sensing materials also possess structural functions and are fabricated with conventional structural materials (e.g., asphalt-based or cement-based materials). Therefore, excellent compatibility with the host material, durability, and mechanical strength are guaranteed. Moreover, the sensitivity of self-sensing materials can be orders of magnitude higher than that of conventional sensors [26].

The first research on the self-sensing behavior of cementitious composites dates back 1993 [27]. Since then, several researchers have highlighted the possibility of using this kind of smart material for monitoring transport infrastructures [23,28,29]. Although the self-sensing behavior of cementitious materials has been widely investigated, in regard to which several reviews can be found in the literature [22,24,30,31,32,33], research on the self-sensing performance of asphalt-based materials is very limited. The experience accumulated over the years on the self-sensing properties of cement-based materials is undoubtedly valuable and can be a source of inspiration for the development of self-sensing asphalt-based materials. However, it should be highlighted that the mechanical behaviors of cement-based and asphalt-based materials are different, thus leading to the necessity of developing tailored studies for the characterization of the self-sensing responses of asphalt materials. Cement-based materials, such as concrete, exhibit linear elastic behavior under loading. On the other hand, an asphalt mixture is a viscoelastic material, meaning that it exhibits both elastic and viscous characteristics; thus, the corresponding mechanical and self-sensing responses strongly depend on both temperature and the time of loading (i.e., frequency). In addition, the fabrication procedure, the self-sensing mechanism, and the laboratory assessment of asphalt-based materials would be obviously different.

To the best of the authors’ knowledge, a review on the self-sensing technology applied to asphalt pavements is still lacking. Thus, the objective of the present review is to fill this gap and lay the groundwork for the development of self-sensing asphalt pavements. It is believed that the subjects addressed in the present review can be helpful for researchers and professionals involved in the road-paving industry who want to approach this innovative line of investigation. 

## 2. Composition of Self-Sensing Asphalt Pavements

Self-sensing asphalt mixtures are multi-phase materials, consisting of a conductive phase (i.e., functional filler), non-conductive phase (i.e., aggregates and bitumen), and the interfaces between the conductive and non-conductive phases, and this characteristic strongly affects the self-sensing performance of a composite [22]. As indicated in Section 1, the self-sensing mechanism is based on changes in the electrical response of an asphalt mixture when subjected to external stimuli. Therefore, the enhancement of the electrical conductivity of asphalt mixtures is an essential prerequisite for endowing this material with a self-sensing function. In this section, the conductive mechanism, the most-used additives, and the dispersion technique for obtaining self-sensing asphalt mixtures are described.

### 2.1. Conductive Mechanism of Asphalt Mixtures

Conventional asphalt mixtures, made of aggregates and bitumen, can be considered insulators [34], with electrical resistance varying from 10^8^ to 10^12^ Ω·m [34]. However, it is theoretically feasible to enhance the conductivity of asphalt mixtures by adding a certain amount of conductive additive [35]. The electrical conductivity behavior of composites is governed by various mechanisms, such as contacting conduction, the tunnelling effect, the field emission effect, and ionic conduction [36]. Contacting conduction occurs due to direct contact between conductive particles, thus forming continuous conductive paths. The tunneling effect is a quantum-mechanical effect that takes place when disconnected particles are close enough, in the order of nanometers [37], to allow electrons to penetrate the energy barrier [22]. The conductive behavior of asphalt mixtures is typically attributed to the contact and tunnelling effects [38]. Field emission conduction is induced by a local strong electric field. According to Han et al. [39], some conductive additives with a unique morphology (e.g., carbon nanotubes) can induce a localized increase in an electric field at sharp tips, promoting field emission conduction. Despite some research showing the effect of field emission on the conduction mechanism of cement-based materials [40], no studies on asphalt-based materials have been conducted. Ionic conduction takes place in the presence of moisture in a matrix due to the motion of ions. Laboratory investigations on the conductive behavior of asphalt-based materials are typically carried out in a dry state (a water-free matrix); hence, the effect of ionic conductivity has not been assessed yet [41]. However, considering that asphalt pavements in service can be subjected to moisture, it is believed that the effect of ionic conduction should be assessed in future research.

The amount of additive strongly affects the electrical conduction properties of asphalt mixtures, allowing the transition from an insulator to a conductor material. Percolation theory has been used to describe the insulating–conductive transition of asphalt [42,43]. A schematic representation of this process is shown in Figure 1. According to percolation theory, the change in the electrical resistivity of asphalt mixtures caused by the incorporation of conductive particles can be divided into four phases [43]. In the first phase, when the additive volume fraction is low, the electrical resistivity of the asphalt mixture slightly decreases. This is because the particles are distributed homogeneously in the volume of the insulating host matrix; hence, no or only few conductive paths are formed, as there is no contact between adjacent particles. When the additive content increases, the transition phase occurs. The particles start to form conductive paths, mainly due to tunneling conduction and/or field emission conduction [44]. There is a critical content, called the percolation threshold, at which point the electrical resistivity dramatically decreases by several orders of magnitude until reaching the conductive phase. Beyond the percolation threshold, the formation of new conductive paths tends to saturate due to contact conduction, and then the electrical resistivity of the mixture does not decrease any further. 

The investigation of the percolation behavior of a conductive asphalt mixture is crucial for proving the technical and economic viability of self-sensing asphalt pavements, as it permits the selection of the optimum content of additive to be incorporated into the asphalt mixture to make it conductive. A minimum amount of conductive additive must be incorporated to form continuous conductive paths. However, using excessive amounts of admixtures can be counterproductive, as this tends to saturate the conductive paths and limit sensing effectiveness [44]. In addition, it increases the production cost of the asphalt material and can lead to the degradation of some volumetric and mechanical properties of the asphalt mixture [35].

### 2.2. Conductive Additives

The most widely used additives in conductive asphalt mixtures are carbon fibers (CF) [35,41,45], graphite powder (GP) [35,41,43,45,46], carbon black (CB) [35], steel slag (SS) [41], steel fibers (SF) [43,46,47,48], and different kinds of carbon-based nanomaterials, such as carbon nanofibers (CNFs), carbon nanotubes (CNTs), graphene nanoplatelets (GNPs) [49,50], and graphene oxide (GO) [51]. 

According to their morphologies, there are two main categories of additives: particles and fibers. Various studies have shown that fibers are preferred for enhancing the conductivity of asphalt mixtures. The higher aspect ratio of fibers promotes the formation of longer conductive paths compared with those of particle additives [41]. Particle additives are easily wrapped by the surrounding matrix phase (i.e., bitumen), and the electrical channels are more likely to be blocked [24]. For this reason, a higher quantity of particles is necessary to enhance the conductivity of the mixture. 

However, a combination of both conductive particles and fibers was found to be the best solution for enhancing the conductivity of a mixture. The addition of CF and GP led to superior performance with respect to single additives [35,41]. Similar results were obtained by the other authors of [43,46], who enhanced the conductivity of asphalt mixtures by adding SF and GP. It was reported that the improvement obtained by the combination of different types of additives was due to the fact that conductive particles exhibit short-range contacts in the form of clusters, while fibers promote a bridging effect between these clusters, leading to the formation of conductive paths [35] (Figure 2).

Wu et al. [35] investigated the percolation behavior of asphalt mixtures containing CB, GP, and CF. The authors found the percolation threshold to be at approximately 16%, 15%, and 6% (by volume of bitumen) for CB, GP, and CF, respectively (Figure 3). The authors also found that different combinations of additives, i.e., GP + CF and CB + CF, strongly enhanced the conductivity of the mixture. It is also worth noting that asphalt mixtures with the same content but different additive combinations exhibited similar conductivity. The data obtained aligned with the trend line of the CF percolation curve. 

In another study, Arabzadeh et al. [41] found that the percolation threshold of CF asphalt mastic was 0.75% by volume of the mastic portion. Wang et al. [46] compared the performance of SF and GP to enhance the conductive properties of an asphalt mixture. According to the results, SF mixtures showed superior performance, as the percolation threshold was reached at a content of 1.72% by volume of binder. In comparison, a total GP proportion of 18% is necessary for achieving low resistivity. Hosseinian et al. [47] added SF to an asphalt mixture and found that the percolation threshold was reached at a content of 6% by weight of bitumen. García et al. [43] also found that SF had the best conductive performance of asphalt mortars when compared with GP. They obtained a percolation threshold at an optimum SF content of 6.02% by volume of bitumen, while the influence of GP was very weak. Wu et al. [52] evaluated the addition of multiple electrically conductive materials, such as GP, CF, SF, and SS. The research results showed that the combination of different additives (fibers and particles) induced the best enhancement in the electrical properties of asphalt mixtures. 

Table 1 summarizes the results obtained by different authors using conductive additives. It seems that the addition of SF is among the best solutions for boosting the electrical properties of asphalt mixtures, as low electrical resistivity can be obtained by adding a low quantity of fibers. However, some issues, like the formation of clusters and issues regarding the mechanical properties of a mixture (see Section 2.3), should be considered. On the other hand, it seems that the employment of carbon-based nanomaterials, such as GNPs, CNFs, and CNTs, for enhancing the electrical properties of asphalt mixtures has been scarcely investigated. Considering the excellent results obtained in the field of self-sensing cement-based materials incorporating carbon-based nanomaterials [32], further research is needed in the field of asphalt-based materials. Furthermore, it is noteworthy that the benefits of using carbon-based nanomaterials with regard to the mechanical properties of asphalt mixtures have been widely assessed [53,54].

### 2.3. Dispersion Techniques

Proper dispersion of conductive additives into an asphalt matrix is crucial for enhancing the electric properties of mixtures and activating their self-sensing function. The most adequate dispersion technique depends on several factors, such as the morphology and the dimensions of the conductive additives. 

For example, steel slag, due to its granular shape, does not necessitate any specific dispersion technique and can be treated as a natural aggregate in an asphalt mixture. Although this is a huge advantage for its employment, it should be noted that the use of only steel slag hardly permits the activation of a self-sensing function; thus, a combination with conductive nanoparticles is often necessary [67]. Steel fibers are typically mixed with bitumen before the incorporation of aggregates [43,68], without any specific modifications in the fabrication phase. However, when higher fiber proportions were used, agglomeration problems were observed [43,69,70]. 

Dispersion is a crucial issue when nanoparticles are incorporated into asphalt materials, as nanoparticles tend to agglomerate due to the corresponding strong van der Waals interactions [71]. Improper dispersion can lead to nanoparticle damage and size break down, deteriorating material properties [72]. To facilitate the dispersion of nanoparticles inside the asphalt binder, different methodologies have been used. Dry methods entail adding the nanoparticles directly into the bitumen without any pre-treatment, and dispersion is achieved via hand-mixing [61] or by means of a high-shear mixer [73]. This is a simple laboratory procedure that can easily be transferred to the industrial scale [74]. On the other hand, the wet (or solvent-blending) method is believed to enhance dispersion performance, but the procedure is more complex [75]. In this technique, nanoparticles are first dispersed in a specific solvent (typically kerosene, acetone, or deionized water) by means of ultrasonication, which permits the disaggregation of individual nanoparticles. In this process, the use of different kinds of surfactants (anionic, cationic, or nonionic) is fundamental for obtaining a homogeneous and stable solution. The process is illustrated in Figure 4. After ultrasonication treatment, nanoparticle agglomerations are dispersed into individual nanoparticles surrounded by surfactant molecules. Various techniques have been used to assess the nanoparticle dispersions in aqueous suspensions, such as scanning electron microscopy (SEM) [76], digital microscopy [71], and UV-Visible Spectroscopy [77]. Moreover, ref. [78] reported that the final electrical resistance of the composites can be used as an indirect indicator of dispersion quality.

Once a homogeneous solution is obtained, it must be incorporated into the hot asphalt binder and mixed with high-shear mixers until the solvent is evaporated and modified bitumen is obtained. The wet method was found to give rise to better dispersion performance than the dry method [53,79]. However, the wet process is expensive and complex; thus, the scaling of this technique is challenging. Table 2 summarizes the most-used procedures for the dispersion of carbon-based nanomaterials into bitumen.

### 2.4. Electrical Measurements

The measurement of the electrical resistance of an asphalt mixture can be carried out using the two-probe and four-probe methods, as shown in Figure 5.

In the two-probe method (Figure 5a), the specimen is placed between two electrodes, used for both current passing (I) and voltage (V) measurement. This is a straightforward technique, as no modifications of the sample fabrication procedure are necessary. Norambuena-Contreras et al. [85] placed two stainless-steel electrode plates on opposite faces of an asphalt mixture sample, and, in order to guarantee good contact between the plates and the sample surface, a low pressure of 1 kPa was applied. Other researchers applied silver paint and graphite powder at the interface between the electrode and the specimens to ensure good contact [35,43,46]. This methodology can be used to determine the electrical resistance of unloaded specimens. However, the assessment of the self-sensing properties of a mixture requires the performance of electromechanical tests (see Section 3), in which mechanical load and electrical measurements are carried out simultaneously. Consequently, the electrodes must be fixed or attached to the specimen. Arabzadeh et al. [41] took advantage of the adhesive properties of bitumen to attach electrodes for two-probe electrical measurements. The contact area of the specimen was slightly warmed, and the electrodes were forced against it, permitting the bitumen to flow and ensure perfect contact (Figure 6a). Notani et al. [64] used copper foils that completely adhered to the specimens after the compaction of the mixture to ensure good contact (Figure 6b). Rew et al. [59] used copper tape attached to the mixture as electrodes. The main shortcoming of the two-probe method is that the measured resistance includes the contact resistance at each of the two electrodes, which, in many cases, is not negligible [86].

To overcome this limitation, the four-probe method should be applied (Figure 5b). In the four-probe method, four electrodes are used for electrical measurement. The current is applied between the two outer electrodes, and the voltage is measured between the two inner electrodes. The main advantage of this technique is that it permits the elimination of the contact resistance between electrodes and the material [87]. Both attached and embedded electrodes can be used to carry out electrical measurement with the four-probe method. However, the embedment of the electrodes is recommended, as this allows better contact and avoids the debonding of the electrodes from the specimen [22]. Although electrode embedment is a relatively straightforward procedure in materials such as concrete, for which many studies are available [88,89], there is no consensus on the most suitable methodology for embedding electrodes in an asphalt mixture. The main issue in this regard is that the fabrication of asphalt mixture specimens requires a step consisting of compaction into cylindrical molds, making electrode embedment very challenging. Rizvi et al. [49] found an effective technique for embedding copper wires electrodes into an asphalt mixture for four-probe electrical measurement. The authors stated that copper wires, compared with copper plates, provide a smoother and better electrical signal. To build a good test specimen, a loose asphalt mixture was poured into a compaction mold in five layers to permit the positioning of the four electrodes. After pouring the material for the first layer, five gyrations of compaction were carried out, and then the electrode was manually placed. The same procedure was used for the next layers. This methodology allows for the placement of the four electrodes into the mixture in an equally spaced fashion. Gulisano et al. [67,90] employed a similar methodology for fabricating Marshall specimens with four embedded electrodes, as shown in Figure 6c. In another study, Gulisano et al. [91] employed a roller compactor for manufacturing prismatic self-sensing asphalt specimens with four copper wire electrodes embedded (Figure 6d). However, a standardized laboratory procedure for the embedment of electrodes inside asphalt mixture is still lacking.

**Figure 6 sensors-24-00792-f006:**
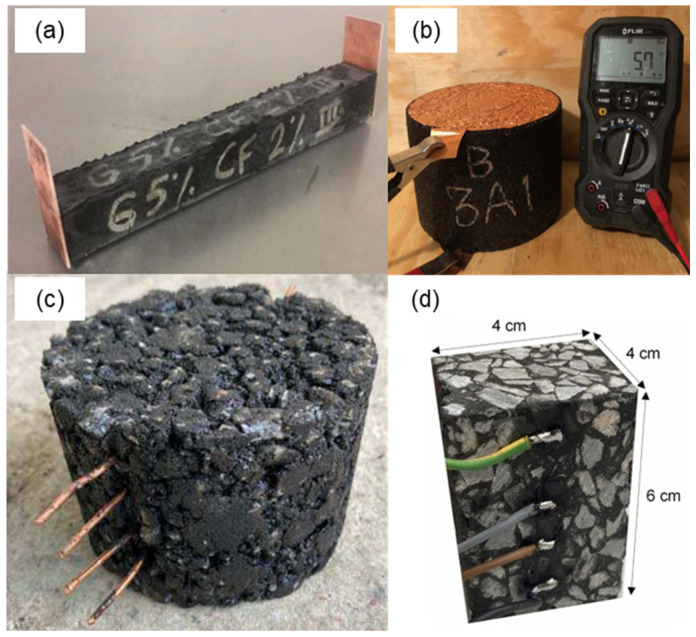
Electrodes’ embedment procedures. (**a**) attached copper plates, (**b**) copper foils, (**c**,**d**) copper wires. Reproduced with permission from [41,64,90,91], published by Elsevier, American Society of Civil Engineers, and Springer Nature.

In measuring electrical resistance using either the two-probe or four-probe methods, the measurement can be conducted with direct current (DC), in which a fixed DC voltage is applied to the specimen, or alternative current (AC), wherein a low-frequency voltage is used. The DC approach is the most-used method for its simpler procedure. In the case of cement-based materials, it was reported that one of the main drawbacks of using the DC method is that it can cause the electrical polarization of a material, which complicates resistivity measurement and is undesirable for the practical implementation of resistivity measurement in the field [87]. A practical method for reducing the polarization effect is to employ alternative current (AC) [92,93]. Other works [94] have reported that the linear relationship between current and voltage reflects the fact that contacting conduction is the dominant conduction process inside the specimen. Conversely, when the tunnelling effect is the dominant conduction process, a nonlinear power function of current and voltage would be induced. Although these processes have been studied for cement-based materials [87], no studies on asphalt-based materials have been found in the literature.

## 3. Self-Sensing Mechanism and Assessment

### 3.1. Piezoresistivity

Piezoresistivity is the most explored sensing function and consists of a reversible change in the electrical resistance of a material when subjected to strain. According to the second Ohm’s law, electrical resistance is given by Equation (1):(1)$$R=\rho \cdot \frac{l}{A}$$
where *ρ* is the electrical resistivity, *l* is the length of the material being tested, and *A* is the cross-sectional area. From Equation (1), the Fractional Change in Resistance (FCR) can be determined as follows [30] (Equation (2)):(2)$$\mathrm{FCR}=\frac{\Delta R}{R_{0}}=\frac{\Delta \rho }{\rho_{0}}+\frac{\Delta l}{l_{0}}\cdot \left(1+2\nu \right)=\frac{\Delta \rho }{\rho_{0}}+\varepsilon \cdot \left(1+2\nu \right)$$
where *ν* is the Poisson’s ratio, and Δ*l*/$l_{0}$ is the strain (*ε*). Piezoresistive response sensibility is typically evaluated with the Gauge Factor (GF), defined as the FCR per unit strain [95] (Equation (3)):(3)GF=FCRε=Δρρ0ε+1+2ν

From Equation (3), it is evident that the piezoresistive sensing effectiveness depends on the contribution of two terms [96]. The first term, (Δ*ρ*/$\rho_{0}$)/*ε*, is an intrinsic electro-mechanical term [97] relating to changes in the distance between adjacent particles and interfacial properties. The second term, (1 + 2*ν*), is a geometrical term, depending on the Poisson’s ratio. Considering that asphalt mixtures have a Poisson’s ratio near 0.35, a gauge factors smaller than 1.7 can be attributed to geometrical contributions, whereas intrinsic electro-mechanical contributions will lead to gauge factors greater than 1.7.

Similarly, stress sensitivity (SS) or load sensitivity (LS) can be determined by dividing the FCR by the applied stress, σ, and load, P (Equations (4) and (5)):(4)$$\mathrm{SS}=\frac{\mathrm{FCR}}{\sigma }$$
(5)$$\mathrm{LS}=\frac{\mathrm{FCR}}{P}$$

Wu et al. [98] subjected a graphite-modified asphalt mixture to a cyclic compressive test with stress amplitudes ranging between 0.7 and 1.2 MPa. According to the authors, the variation in the electrical resistance of conductive asphalt-based materials when subjected to stress/strain may involve a proximity effect, microcracks, and the dislocation of conductive paths due to the shear motion of aggregates. The proximity effect refers to the fact that, during tensile and compressive loading, the conductive particles tend to separate and approach, respectively, leading to changes in electrical resistance. Furthermore, the formation of microcracks during loading cuts some conductive paths, increasing electrical resistance. Further variations are due to the shear motion of aggregates and viscous behavior, leading to permanent deformation and unrecoverable electrical resistance [49]. Liu et al. [55] studied the GF of conductive asphalt mixtures containing graphite and carbon fibers. For a stress amplitude of 1.5 MPa, the GF of the conductive asphalt mixture during an indirect tensile test with a load frequency of 1 Hz was 11 ± 2. In another study, Liu et al. [45] tested the strain-sensing properties of an asphalt mixture consisting of 15% graphite and 2% carbon fibers by volume of bitumen. The authors subjected the specimens to various cycles of indirect tensile loading/unloading tests. Each cycle had a duration of approximately 300 s. When applying a stress amplitude of 0.7 MPa, the GF was 350 ± 50. In order to better simulate the behavior of asphalt mixtures in situ, Rizvi et al. [49] subjected asphalt mixtures with CNFs to sinusoidal, haversine, and creep loading. Some of the results obtained are illustrated in Figure 7, showing the excellent sensing response of this kind of mixture. Gulisano et al. [67,91] subjected EAFS and GNPs asphalt mixtures to loading at different stress levels, showing that these kinds of mixtures exhibited stress-sensing capabilities (Figure 8). 

To use self-sensing pavement for traffic monitoring or WIM applications, good sensitivity properties must be guaranteed during the lifespan of the asphalt pavement. Liu et al. [99] subjected asphalt mixtures with different graphite fractions (35 wt% and 40 wt% of the bitumen) to ten loading/unloading compressive cycles, the results of which are shown in Figure 9. The fractional change in resistance (FCR) decreased after each cycle for both mixtures, probably due to permanent deformations. The GF of the mixture with higher content of graphite (Figure 9b) decreased with the increase in the number of cycles, indicating a reduction in sensitivity due to the deterioration of the specimen. On the other hand, the mixture with lower graphite content seemed to maintain the same GF after ten loading/unloading cycles. This result suggests that the long-term stability of piezoresistive properties may depend on the conductive properties of the mixture. Future research should evaluate the effect of the additive content on the long-term stability performance of asphalt mixtures. Gulisano et al. [100] employed a digital-signal-processing (DSP) algorithm and a dense Artificial Neural Network (ANN) model to develop a load classification using electrical data generated by a piezoresistive carbon fiber (CF) asphalt mixture. The model demonstrated a test set accuracy of 0.977, showing sensing capabilities and indicating the potential application of these materials in weigh-in-motion systems.

### 3.2. Damage Sensing

Self-sensing materials also change their electrical resistivity when there is damage in their structures. Although this mechanism is different from strain sensing and cannot be strictly attributed to piezoresistivity [30], damage sensing is an essential function for early damage detection and pavement-health-monitoring operations. The formation and propagation of microcracks inside a material cause the cutting of some conductive paths, which lead to a gradual increase in electrical resistance. The continuous monitoring of the electrical resistance of asphalt pavement would therefore permit the early detection of pavement distress and allow the implementation of preventive/predictive maintenance operations.

The ability of an asphalt material to sense its damage condition is typically evaluated with mechanical tests conducted until the failure of the specimen. Static and dynamic (e.g., fatigue) tests can be conducted for this purpose. An asphalt mixture will exhibit good damage-sensing properties if a predictable relationship between the FCR and the structural conditions of the pavement can be obtained.

Liu et al. [99] evaluated the damage-sensing properties of asphalt mixtures with graphite and carbon fibers through a cyclic compression fatigue test conducted until the failure of the specimens. According to the results of this study, the FCR during the fatigue test can be divided into three phases (Figure 10). In the first phase, cyclic loading caused further compaction of the specimen, which reduced the tunnelling distance and led to a reduction in electrical resistance. In the second phase, the additional compaction and the formation of microcracks in the specimen led to the continuous formation and destruction of conductive paths, resulting in an equilibrium state in the FCR. Finally, in the third phase, crack propagation dominated the conductive mechanism, causing the destruction of the conductive network and a sharp increase in the FCR until the failure of the specimen. Similar trends were obtained by Wu et al. [56], who tested an asphalt mixture containing graphite powder in a cyclic indirect tensile test until its failure.

In another study, Liu et al. [101] subjected a graphite-and-carbon fiber asphalt mixture to a static tensile test conducted up to the failure of the specimen. A good correlation between longitudinal displacement and electrical resistance was observed. Firstly, the electrical resistance slightly decreased. Then, the electrical resistance gradually increased, and, finally, it rose abruptly before the failure of the specimen. In another work, Liu et al. [102] identified a correlation between the resistivity of carbon fiber graphite asphalt concrete and its degree of cracking. Their resistance increases with the widening of cracks, and once cracking reaches a specific threshold, there is a sudden change in resistance, signifying the destruction of the mixture. Rizvi et al. [49] designed asphalt mixtures containing 6.5% carbon nanofibers by weight of binder and tested their damage-sensing properties through a compressive loading test. A more complex behavior was observed during static compressive loading, and various stages of increase/decrease in FCR were observed. Gulisano et al. [67] showed that asphalt mixtures incorporating Electric Arc Furnace Slag (EAFS) as fine aggregates and GNPs (7% by weight of the binder) showed an excellent damage-sensing response, and their electrical resistance gradually increased during a compression test. Ullah et al. [65] also found a good relationship between electrical resistance and crack formation/propagation in asphalt mixtures with carbon fiber (CF) and iron tailings aggregates (TA) under fatigue tests. Gulisano et al. [91] employed a digital-signal-processing technique based on the continuous wavelet transform method and discrete wavelet multiresolution analysis to assess the electrical signal generated by a self-sensing asphalt mixture in the time–frequency domain. The results of the study (Figure 11) showed that this kind of technique can capture valuable information about the formation and propagation of cracks inside a specimen.

The results of the aforementioned studies show that although conductive asphalt mixtures exhibit damage-sensing properties, the proper control of this function has not yet been achieved. Statistical models aimed at studying the relationship between the FCR and pavement structural conditions must be developed in the future. Such analysis would permit the treatment of the FCR as a pavement condition index and the development of continuous pavement-health-monitoring systems. 

### 3.3. Temperature and Moisture Sensing

Conductive asphalt mixtures can also exhibit temperature- and moisture-sensing properties. Real-time monitoring of the temperature of pavement is an important task, which is typically carried out with embedded thermal devices (e.g., thermocouples, thermistors, etc.) [5]. The development of intrinsic thermoresistive asphalt mixtures would negate the need for the employment of embedded thermal devices. In addition, knowledge of the relationship between electrical resistance and temperature is also necessary for negating the effect of temperature during piezoresistive and damage-sensing measurements. The change in electrical resistance caused by the influence of the change in the surrounding temperature is known as the thermoresistive effect [103]. The thermoresistive effect arises because temperature affects the number of carriers and their mobility in a composite material. Thermoresistive sensitivity can be assessed by using the Temperature Coefficient of Resistance (TCR), defined as follows (Equation (6)):(6)TCR%°C=ΔRR0·1ΔT·100=FCRΔT·100

Wu et al. [52] found that conductive asphalt mixtures exhibited a Positive Temperature Coefficient (PTC), meaning that electrical resistance increased as the temperature increased. According to the authors, this behavior may be due to the expansion of the bitumen, causing separation among conductive particles and an increase in electrical resistance. PCT behavior was also obtained by Sun et al. [60] with carbon black asphalt mixtures. However, other studies performed on cementitious materials with a higher coefficient of thermal expansion than asphalt mixtures showed different behaviors. Wen et al. [104] found that carbon-fiber-reinforced cement exhibited a Negative Temperature Coefficient (NTC), meaning that an increase in temperature caused a reduction in electrical resistance. NTC behavior is due to the fact that a temperature increase induces a decrease in the tunnelling distance between conductive particles due to thermal fluctuations, resulting in a reduction in the electrical resistance of composites [105,106]. In another study, Chen et al. [107] observed both PCT and NTC behavior in carbon-fiber-reinforced concrete, depending on the temperature range. To sum up, the thermoresistive behavior of conductive asphalt mixtures is still uncertain, and further research is needed.

The presence of moisture can weaken asphalt–aggregate adhesion and result in aggregate stripping and moisture damage in an asphalt mixture [108]. Ground-Penetrating Radar (GPR) [109] can be used for the detection of moisture content in asphalt pavements [110,111]. However, the use of asphalt mixtures with moisture-sensing functions would permit the continuous monitoring of moisture content. Although moisture sensors based on nanomodified Portland cement have been designed [112], the moisture-sensing function of asphalt materials has not yet been assessed.

## 4. Full-Scale Implementation

The previous sections outlined the fundamental aspects of asphalt-based self-sensing materials, including their compositions, their fabrication procedures, and laboratory-scale assessments of their sensing performance. While the results obtained so far are promising, it is important to note that the technical maturity of this technology is still low, and further research is required before it can be scaled up for real-world applications. Nonetheless, in recent years, some researchers have explored the possibility of implementing self-sensing road pavements in full-scale applications, leading to a few real-life-scale studies. Consequently, this section aims to present an overview of the studies conducted thus far and establish a foundation for future full-scale investigations.

The concept behind self-sensing technology involves incorporating self-sensing asphalt materials into specific sections of roadways [67], as depicted in Figure 12. The electrical data collected from these sections are transmitted to a data center, where they are filtered and processed for purposes such as traffic monitoring and structural health monitoring.

Various self-sensing configurations can be employed, such as bulk or array arrangements [31]. In the bulk configuration, the entire section is constructed using self-sensing materials, requiring the placement of wire electrodes before the fabrication of the pavement. From a construction standpoint, this configuration appears to be the most practical [22]. As shown in Figure 13a, Birgin et al. [113] used a wooden frame to install 3 m long copper wires with a 1 mm diameter in a road section. The electrodes were embedded into a self-sensing smart composite material composed of a thermoplastic binder, short carbon microfibers, and natural aggregates. Conversely, in the array arrangement (Figure 13b), small self-sensing sensors were prefabricated and embedded into the road pavement. Compared to the bulk configuration, the array arrangement reduces development costs by requiring the use of less conductive filler. However, the distinct mechanical responses (e.g., elastic modulus) between the sensor and the host material can limit the accuracy of strain measurements [114]. It is worth noting that the array arrangement has primarily been utilized for cement-based self-sensing materials, and no studies have been conducted on asphalt-based materials. Given the extreme temperatures and mechanical conditions to which the sensors would be subjected during the compaction of asphalt pavement, further studies are necessary in order to validate the performance stability of the small, prefabricated sensors embedded in the pavement.

Once the pavement has been constructed, the electrodes must be connected to a specific electronic circuit and an ad hoc data acquisition system (DAQ). Han et al. [115] used a circuit configuration in which an external voltage input of 12 V was applied to array-arranged cement-based sensors. Each sensor was series-connected with a constant reference resistance, and an A/D card was used to collect the voltage signals, as shown in Figure 14a. Birgin et al. [116] proposed a low-cost DAQ for developing a weigh-in-motion (WIM) system incorporated in a smart pavement, consisting of a control unit, a camera, a regular 5 V USB charger, a voltage regulator, and an analog-to-digital converter, that is controlled by an Arduino Nano microcontroller board (Figure 14b). In another study, the same authors [116] developed a DAQ system for WIM applications consisting of a voltage reader, a micro-controller, a battery, and a USB data output unit. 

The electrical data collected by self-sensing road pavements should be adequately processed for the development of different applications, including those related to traffic monitoring, weigh-in-motion, and pavement health monitoring. 

According to Han et al. [117], when a vehicle passes over self-sensing pavement, the electrical resistance of the material changes (i.e., the piezoresistive effect). By measuring the electrical fluctuations in real time, the traffic volume can be detected. Furthermore, if a pair of piezoresistive sensors are placed at a fixed distance, the speed of the vehicles can also be estimated. A series of studies performed at the outdoor research laboratory of the University of Minnesota Duluth demonstrated that self-sensing technology can be used for road-traffic-monitoring operations. In the first study conducted in 2009, Han et al. [117] embedded a cement-based sensor with CNTs in a concrete road. The sensor was connected to the data acquisition system through wires and electrodes for electrical resistance measurement. A series of vehicles passed over the sensor, and the electrical response was recorded. In a second study, the same authors [115] tested the performance of a nickel-particle-filled cement-based sensor array embedded in road pavement (Figure 15). The results of both studies were promising and showed that self-sensing a CNT cementitious composite could detect traffic flow and even identify different vehicular loadings for weigh-in-motion applications.

Knowledge of the relationship between load and electrical resistance changes (i.e., load sensitivity) can also be used for weigh-in-motion (WIM) applications. As the amplitude of the electrical changes depends on the applied load, the measurement of amplitude would permit one to estimate the weight of vehicles and to obtain the axle load spectra, which are fundamental for pavement design and rehabilitation applications. Birgin et al. [113] used electrical data from a self-sensing pavement to develop a weigh-in-motion (WIM) system. The typical output of the WIM system is shown in Figure 16. The results of the study show that the truck’s axles are clearly visible and that the summation of the amplitudes of the electrical peaks is proportional to the gross weights of the passing vehicles.

In addition, it was reported that conductive asphalt mixtures possess damage-sensing functions (see Section 3.2). Crack formation and the deterioration of pavement lead to the cutting of the conductive networks, resulting in a change in the electrical resistivity of the mixtures. Knowing the relationship between the electrical response and the pavement’s structural condition would permit treating the electrical response as a pavement condition index (PCI). Predictive models could then be designed to estimate the Remaining Service Life (RSL) of the pavement based on the continuous monitoring of the electrical response. Although laboratory research has assessed the damage-sensing properties of conductive asphalt mixtures, no full-scale studies have been realized.

## 5. Current Perspectives

The rapid advancement of innovative technologies in the transportation sector, including connected and autonomous vehicles, alongside the ongoing digitalization of the construction industry, necessitates a corresponding adaptation in the asphalt-paving sector. This adaptation is crucial to ensure that this sector keeps pace with these advancements and enables the seamless integration of these innovations. Additionally, it is believed that the implementation of sensing technologies in the asphalt sector can drive this transition forward. The previous sections provided an overview of the state-of-the-art self-sensing asphalt pavement technology, highlighting its potential. In this section, the main challenges in this area will be discussed to provide a clear understanding of the current state of this technology and its prospects.

The use of smart self-sensing materials is rapidly growing in many different fields, including medical device manufacturing [118], robotics [119], tissue engineering [120], the automotive industry, and aeronautic structure design [121]. One of the primary advantages of these applications is the relatively easy integration of different sensors thanks to the small dimensions of the host materials. However, the integration of sensing technologies in the construction sector can be highly challenging. This challenge is particularly evident in the case of linear infrastructures, such as road pavements that often stretch across thousands of kilometers. 

Many authors agree that the use of self-sensing technologies in the construction sector offers numerous advantages over conventional embedded sensors, primarily due to their compatibility with the host structure. However, the development and integration of these technologies present new challenges that have not been encountered in the asphalt-paving industry.

First, fabricating accurate self-sensing asphalt-based materials poses a major challenge. Achieving the proper dispersion of functional additives, such as nanomaterials, within the mixture can be difficult. The dispersion of nanomaterials, for instance, may require precise dosing, the addition of surfactants, ultrasonication, and tailored procedures for incorporating the nanomaterials into the bitumen. Additionally, it has been observed that the conductivity properties of these materials can be influenced by various factors, including loading frequency, temperature, humidity, and others. Consequently, one of the primary challenges for field applications is to evaluate the reliability and sensitivity of these systems, particularly when exposed to fluctuations in traffic loads and variations in environmental conditions.

Additionally, the lack of standardized procedures for assessing self-sensing properties in a laboratory setting currently hinders the scaling of this technology.

Apart from the fabrication challenges, real-scale application introduces additional obstacles that need to be addressed. Self-sensing asphalt pavement must be effectively connected to a specific data acquisition system for gathering electrical data. However, research in this area is scarce, and efforts should focus on finding effective, easy-to-install, stable, and low-cost systems. Data processing and analysis present additional obstacles for full-scale applications, as most of the research conducted thus far has been carried out in controlled laboratory environments. Considering the complex behavior of asphalt mixtures, which are thermoplastic and viscoelastic materials, data processing can be complicated. Therefore, employing artificial intelligence algorithms capable of analyzing the electrical signals generated by pavement and providing insightful information about its structural condition is crucial. 

Lastly, all the aforementioned factors naturally increase the cost of pavement construction, including costs associated with nanomaterials, their incorporation into the mixture, installation, data acquisition, and analysis. However, as extensively discussed throughout this paper, the development of this technology is believed to yield enormous benefits in the transportation sector. Firstly, the early detection of pavement deterioration would lead to significant savings in resources dedicated to road maintenance, resulting in notable economic savings. Additionally, it would reduce traffic congestion and the associated pollution caused by road rehabilitation works. Moreover, it would enhance road safety by ensuring optimal pavement conditions, reducing the risk of accidents and providing safer driving experiences for road users. Furthermore, it would optimize maintenance strategies, leading to a decrease in the consumption of non-renewable resources such as bitumen and aggregates, by avoiding unnecessary operations.

## 6. Conclusions

The paper presents an overview of the topic of self-sensing asphalt pavements, covering the principles, compositions of the materials, laboratory assessment, and full-scale applications. The main conclusions of this paper can be summarized as follows:The self-sensing mechanism is based on changes in the electrical response of an asphalt mixture when exposed to external stimuli. To enable this function, conductive additives must be properly incorporated into the asphalt mixture. Percolation analysis describes the relationship between additive content and the electrical resistivity of a mixture, representing a fundamental tool for evaluating the technical and economic feasibility of self-sensing asphalt pavements.Conductive additives used for self-sensing purposes typically include metallic or carbon-based materials, such as steel slag, steel fibers, carbon fibers, graphite, and various types of nanomaterials, like graphene, carbon nanofibers, nanotubes, and nanoplatelets. The proper dispersion of conductive additives into the asphalt mixture is crucial for developing self-sensing asphalt-based materials. The most suitable dispersion technique depends on factors such as the morphology and dimensions of the conductive additives. Although some research has been conducted on suitable dispersion techniques, their effect on the self-sensing performance remains unclear.While some laboratory procedures have been employed to assess the self-sensing properties of asphalt mixtures, future standardization is necessary for electrode configuration, sample fabrication, and electromechanical performance evaluation.The current studies have demonstrated the feasibility of incorporating strain-, damage-, and temperature-sensing functions into asphalt mixtures. However, there are still few studies in this area. Further investigations should focus on analyzing self-sensing performance under more realistic dynamic loading conditions and the influence of climatic conditions.Self-sensing asphalt pavements can be utilized in various applications, including those entailing weigh-in-motion, pavement health monitoring, and traffic monitoring. However, the technological maturity of this approach is still low, and additional research is required before it can be implemented on a larger scale for real-world applications. Dedicated efforts should be directed towards refining the installation procedures, developing data acquisition systems, and utilizing artificial intelligence to analyze the electrical signals generated by asphalt mixtures, thus creating valuable tools for the field of transportation engineering.

## Figures and Tables

**Figure 1 sensors-24-00792-f001:**
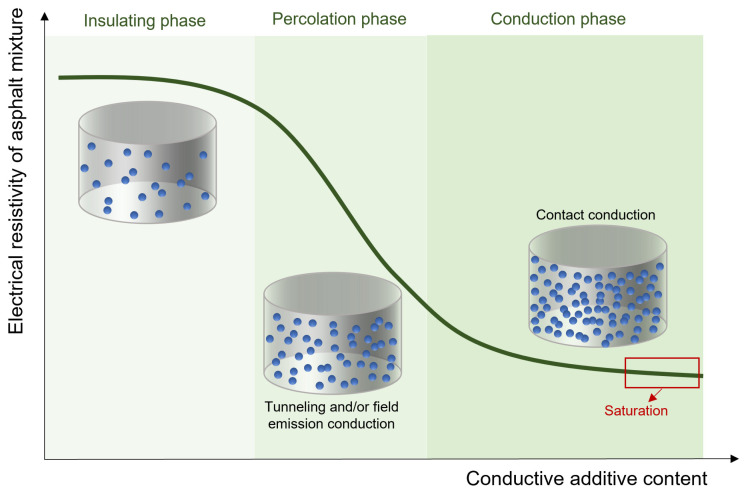
Percolation theory pertaining to asphalt mixtures.

**Figure 2 sensors-24-00792-f002:**
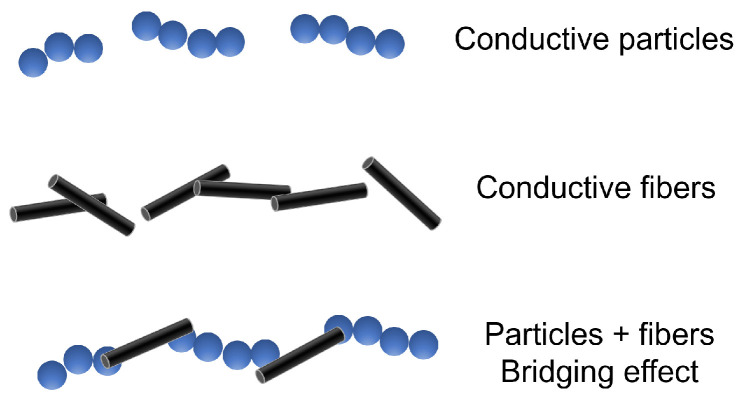
Combining conductive particles and fibers.

**Figure 3 sensors-24-00792-f003:**
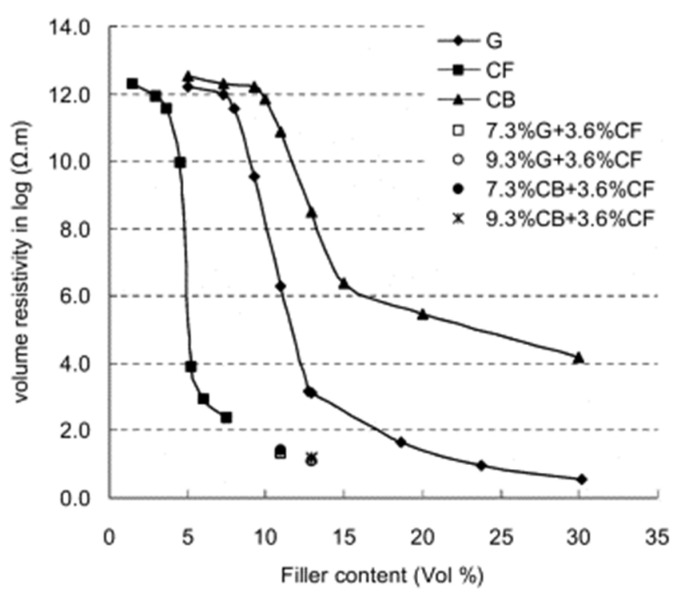
Percolation thresholds of conductive asphalt mixtures. Reproduced with permission from [35], published by Elsevier.

**Figure 4 sensors-24-00792-f004:**
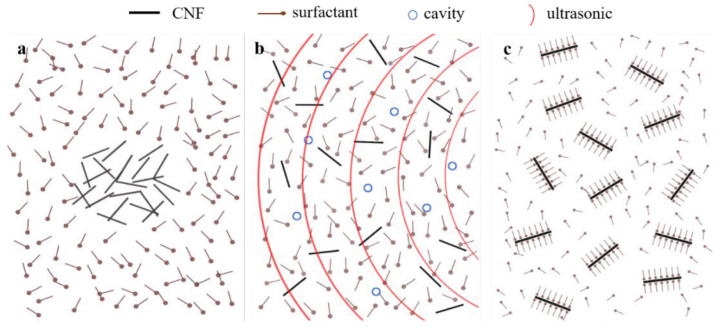
Ultrasonication for the dispersion of nanoparticles: (**a**) before, (**b**) during, and (**c**) after the process [62].

**Figure 5 sensors-24-00792-f005:**
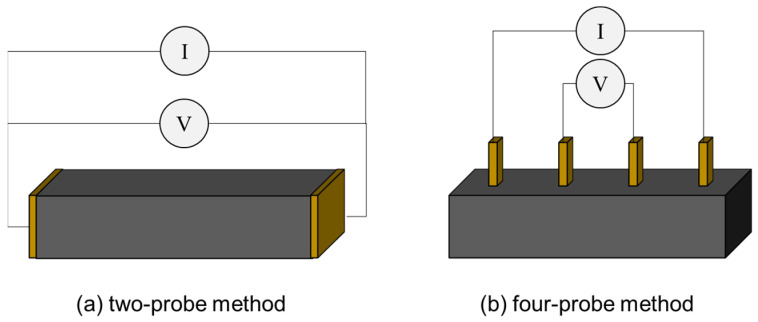
Electrical measurement of asphalt mixtures.

**Figure 7 sensors-24-00792-f007:**
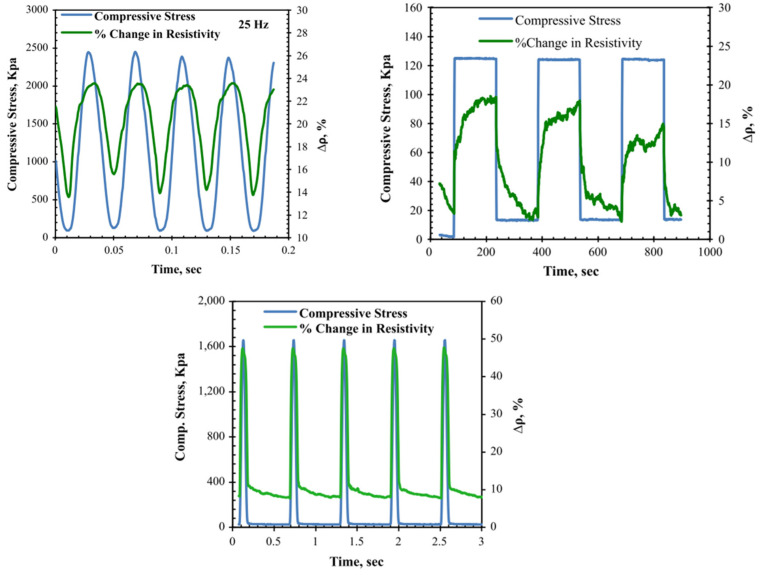
Piezoresistivity of asphalt mixtures under different types of loading. Reproduced with permission from [49], published by Elsevier.

**Figure 8 sensors-24-00792-f008:**
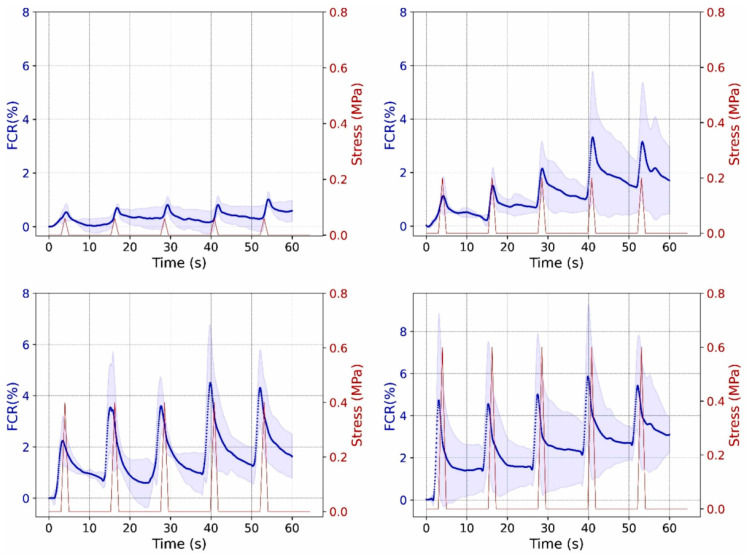
Stress-sensing capabilities of EAFS + GNPs asphalt mixtures [91].

**Figure 9 sensors-24-00792-f009:**
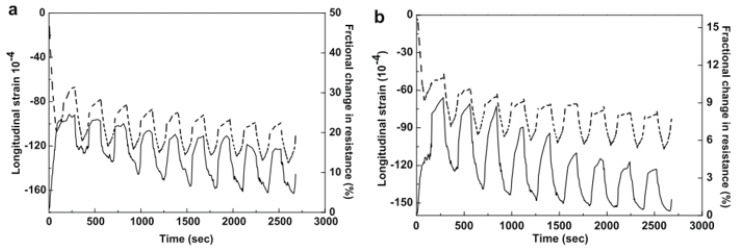
Effect of loading/unloading cycles on piezoresistivity longitudinal strain (dashed curve) and FCR (continuous curve): (**a**) GP content of 35 wt% and (**b**) GP content of 40 wt%. Reproduced with permission from [99], published by Elsevier.

**Figure 10 sensors-24-00792-f010:**
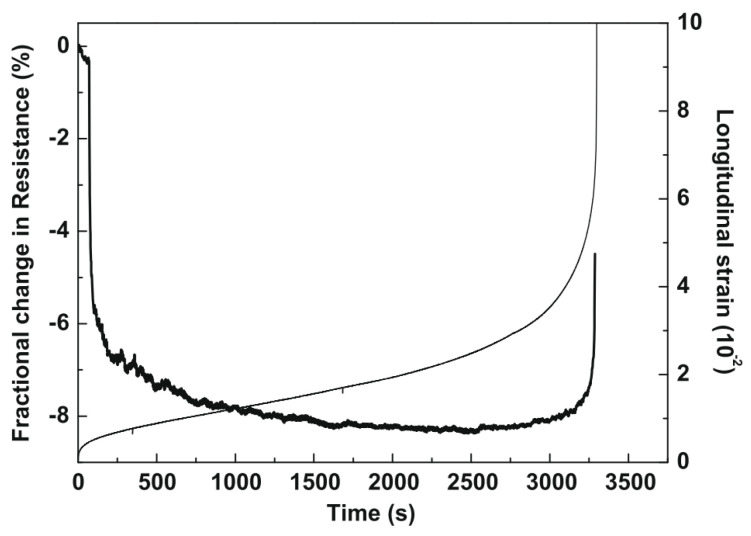
Damage sensing in a fatigue compression test, longitudinal strain (thin curve), and FCR (thick curve). Reproduced with permission from [99], published by Elsevier.

**Figure 11 sensors-24-00792-f011:**
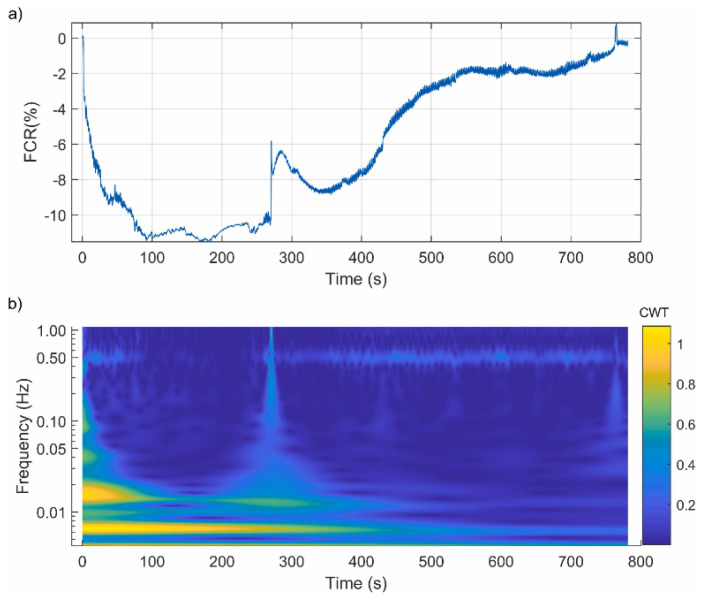
Damage-sensing properties of GNP asphalt mixtures: (**a**) original electric response; (**b**) wavelet assessment [91].

**Figure 12 sensors-24-00792-f012:**
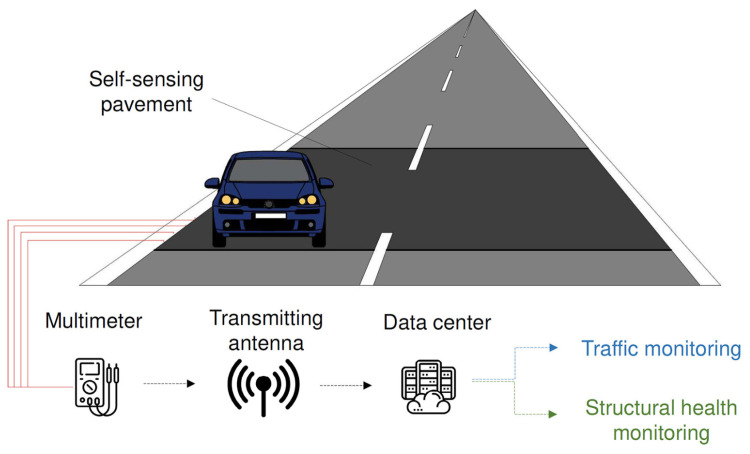
Self-sensing road pavement. Reproduced with permission from [67], published by Elsevier.

**Figure 13 sensors-24-00792-f013:**
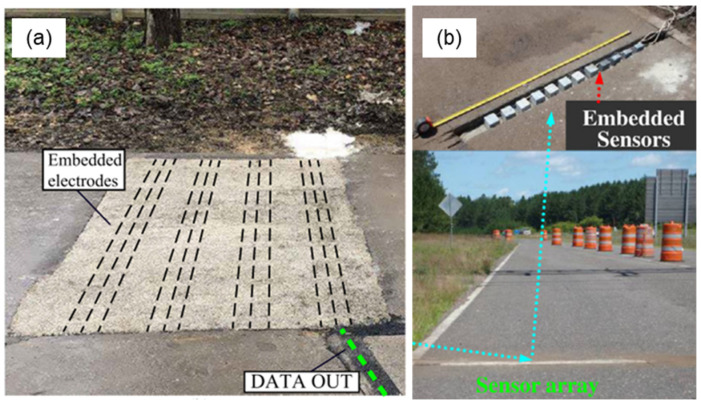
Self-sensing configurations: (**a**) bulk [113] and (**b**) array arrangements (reproduced with permission from [115], published by Elsevier).

**Figure 14 sensors-24-00792-f014:**
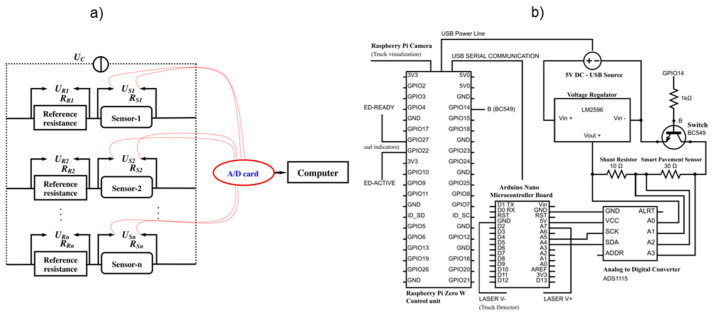
Data acquisition systems for self-sensing pavements: (**a**) reproduced with permission from [115], published by Elsevier, (**b**) [113].

**Figure 15 sensors-24-00792-f015:**
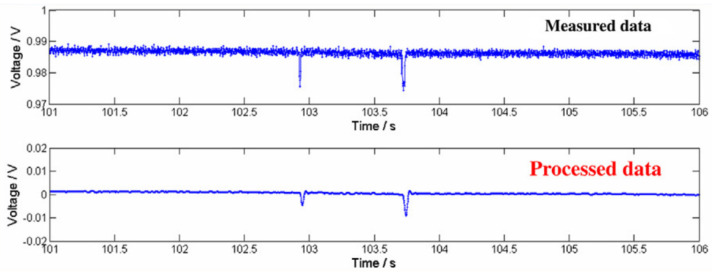
Electrical signals for traffic monitoring. Reproduced with permission from [115], published by Elsevier.

**Figure 16 sensors-24-00792-f016:**
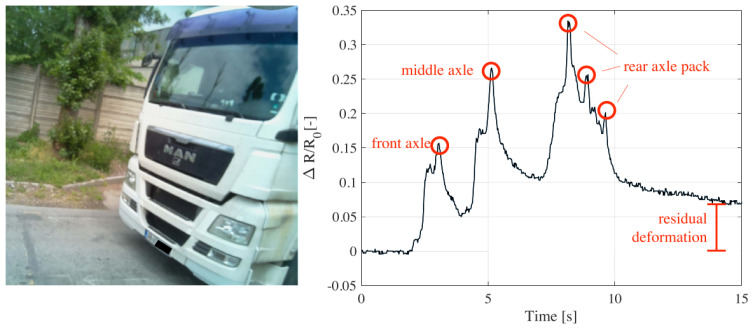
Weigh-in-motion system based on self-sensing technology [113].

**Table 1 sensors-24-00792-t001:** Conductive additives for asphalt mixtures.

Origin	Morphology	Additive	Electrical Resistivity (Ω·m)	Percolation Threshold (% vol of Binder)	References
Carbonaceous	Filler	Graphite powder (GP)	10^−6^–5 × 10^−4^	11–17	[35,41,42,46,52,55,56,57,58,59]
		Carbon black (CB)	10^−4^–8 × 10^−2^	16	[35,60]
		Graphene nanoplatelets (GNPs)	1.25 × 10^−5^	-	[50,61]
	Fiber	Carbon fibers (CFs)	10^−5^–10^−3^	5–14	[52,62,63,64,65]
		Carbon nanofibers (CNFs)	-	-	[49,62]
		Carbon nanotubes/Multi-walled nanotubes (CNTs/MWCNTs)	8 × 10^−6^	-	[66]
Metallic	Filler	Steel slag (SS)	-	-	[50,52]
	Fiber	Steel fibers (SF)	7 × 10^−9^–7 × 10^−7^	3–6	[43,46,52,58]

**Table 2 sensors-24-00792-t002:** Dispersion techniques for nanoparticles.

Dispersion Technique	Nanomaterial	Procedure	References
Dry	CNTs	CNTs were added and manually blended in the bitumen; then, the bitumen–CNT blends were mixed with a mechanical stirrer (1550 rpm for 40 min at 160 °C)	[74]
	GO	High-shear mixing: 4000 rpm for 30–45 min	[51,80,81]
	CNTs	High-shear mixing: 5000 rpm for 30 min	[54]
	GNPs	A glass rod was used to mix GNPs into a binder for 10 min at 150–160 °C	[61,82]
	GNPs	High-shear mixing: 1720 rpm for 1 h at a temperature of 140 °C.	[83]
Wet	CNFs	Solvent: kerosene Sonication time and program: three cycles of 8 min each with an idle (cool-down) time of 25 min between cycles Mixing time: 30 min (120–160 °C), followed by 150 min at 160 °C using low-speed shear mixer.	[53,84]
	CNTs	Sonication time: 60 min High-shear mixing: 1550 rpm for a total time of 90 min at a temperature of 150 °C	[79]
	CNTs	Solvent: methanol Sonication time: 2 h High-shear mixing: 3000 rpm for 45 min (158 ± 5 °C) to ensure the evaporation of methanol	[76]
	GNPs-CNTs	Solvent: deionized water Surfactant: Stock US4498 Sonication time: 30 min	[71]

## Data Availability

Data are contained within the article.

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
