# Peer review of "Development of Self-Sensing Asphalt Pavements: Review and Perspectives"

_sensors, 2024, doi:10.3390/s24030792_

Round 1
Reviewer 1 Report
Comments and Suggestions for Authors
The manuscript reviews the research progress of self-sensing asphalt pavements. The topic is interesting. The research on self-sensing asphalt pavements is just beginning. A review of this topic should not only describe the research progress, but also discuss the method for solving the problems. Some detail comments are shown below:
1. P4, Line 185, The abbreviation for 'graphite powder' should be 'GP' according to the manuscript content and the corresponding references.
2. Please check the data in Fig. 3. The conclusion ‘different combinations of additives strongly enhanced the conductivity of the mixture’ is right. But mixtures with the same content but different additive combinations have the same conductivity, and the data is on the trend line of CF conductivity. This is interesting. It’s recommended to do further analysis using the data in Fig.3.
3. Section 3,It’s recommended to further analyze the reliability and sensitivity of these mechanisms in application, especially when traffic loads and environmental conditions (such as temperature and moisture) change.
4. Section 4, Most of the references are about the full-scale implementation of cement concrete pavement. It’s suggested to slim down the content of this section.
Author Response
First and foremost, the authors would like to express their gratitude to the reviewer for providing valuable comments that enabled us to clarify certain aspects of our paper and enhance its quality. We have diligently worked to address the reviewer's comments, and we believe that the revised paper aligns with the specified requirements:
- Thank you, it has been modified.
-
Thank you for the insightful comment. In the revised paper, a paragraph has been added to highlight this interesting observation:
"It is also worth noting that asphalt mixtures with the same content but different additive combinations exhibit similar conductivity. The data aligns with the trend line of the CF percolation curve."
-
Thank you for addressing this crucial issue. External factors, such as traffic loads and environmental conditions, can influence the sensitivity of these materials. Therefore, prior to the implementation of self-sensing systems on a real-world scale, one of the primary concerns involves studying the impact of these external factors. While the sensing mechanisms are detailed in Section 3 of the paper, it has been decided to include an additional paragraph in Section 5 that more effectively describes the future challenges of these systems:
“One of the primary challenges for field applications would be assessing the reliability and sensitivity of these systems, especially when exposed to variations in traffic loads and changes in environmental conditions (such as temperature and moisture).”
-
Thank you for the feedback. Indeed, numerous full-scale applications of self-sensing road pavements have been implemented with concrete materials. Section 4 of the current paper, focused on full-scale implementation, aims to discuss the primary considerations for the practicality of this technology. This concept encompasses not only the choice of materials (asphalt or concrete) but also other crucial aspects, including electrical measurements, data acquisition systems, etc. While it is true that many of the detailed full-scale experiences involved concrete materials, the underlying concepts can be extrapolated to future studies in asphalt pavements. Therefore, the author believes it is essential to document these full-scale experiences, as they can serve as inspiration for potential future applications involving asphalt materials.
Reviewer 2 Report
Comments and Suggestions for Authors
This paper is in form a review only, but the authors performed a correct and thorough review of the literature in the context of self-sensing asphalt pavements. In the paper is no description of authors experiences in this area. The work would then have a greater response due to the practical presentation of the topic. In comments and perspectives, the authors should pay more attention to the technical aspect of build self-sensing asphalt pavements because this is the biggest problem. The topic is extremely interesting and at the same time difficult due to the large number of factors influencing to obtain a conductive pavement. In my opinion in the part of paper: conclusion and perspective, by authors too little attention was paid to critical technical aspects doing such matterials (obtaining homogeneity, conductivity, performance properties, etc.)
Author Response
First and foremost, the authors would like to express their gratitude to the reviewer for providing valuable comments that enabled us to clarify certain aspects of our paper and enhance its quality. We have diligently worked to address the reviewer's comments, and we believe that the revised paper aligns with the specified requirements.
Thank you for your insightful review of the paper. The authors acknowledge and agree with the review; the topic of self-sensing asphalt pavements is indeed fascinating, but further research is imperative to evaluate the real-scale applicability of these systems. In Section 5, "Current Perspective," we have attempted to summarize the main important issues that urgently need assessment. The dispersion and homogeneity of additives are among these factors, discussed in Section 2.3 and reiterated in Section 5.
In the revised paper, we have included an additional paragraph addressing another aspect related to materials—the impact of external factors on the conductivity and sensing performance of this technology:
“Additionally, it has been observed that the conductivity properties of these materials can be influenced by various factors, including loading frequency, temperature, hu-midity, and more. Consequently, one of the primary challenges for field applications would be to evaluate the reliability and sensitivity of these systems, particularly when exposed to fluctuations in traffic loads and variations in environmental conditions.”
Reviewer 3 Report
Comments and Suggestions for Authors
This is one of the best literature reviews I have ever read. Well done authors! Some minor comments are:
1. Is figure 1 and 2 from any reference? If so, please include citation in the figure name, similar to figure 3 [35]
2. Line 221: “is reached for a content of 6%” should be “is reached at a content of 6%”
3. Line 246-247 is a broken sentence. Change “among which” to “such as” will fix it.
4. Figure 4: add the meaning of a, b, c as notes of the figure.
5. Equation 1,2,3: letter l and number 1 are very confusing. Cannot tell which is which.
6. Figure 9: add the meaning of a, b as notes of the figure.
7. Line 488: “Identified” should not be capitalized.
8. Figure 11: add the meaning of a, b as notes of the figure.
Comments on the Quality of English LanguageIncluded in previous comments.
Author Response
First and foremost, the authors would like to express their gratitude to the reviewer for providing valuable comments that enabled us to clarify certain aspects of our paper and enhance its quality. We have diligently worked to address the reviewer's comments, and we believe that the revised paper aligns with the specified requirements.
- Thank you. Figures 1 and 2 have been realized by the authors of the paper.
-
It has been modified.
-
Thank you, it has been modified.
-
Thank you for the suggestion. Figure 4 shows the different phases of the dispersion process, showing the effect of the ultrasonication. It has been modified the title of the figure: Ultrasonication for the dispersion of nanoparticles, a) before, b) during and c) after the process [62].
-
Thank you, the font of the equations has been now changed to italics, which better permits to distinguish between “l” and “1”.
-
Thank you. The two plots represent different graphite powder (GP) contents. It has been modified the title of the figure. “Figure 9. Effect of loading/unloading cycles on the piezoresistivity longitudinal strain (dashed curve) and FCR (continuous curve) a) GP content 35 wt% and b) GP content 40 wt%.”
-
Ok
-
Thank you. The two figures represent, respectively, the original electrical signal response of the mixture and the wavelet analysis performed to evaluate the damage sensing performance. It has been modified the title of the figure “Figure 11. Damage-sensing of GNPs asphalt mixtures, a) original electric re-sponse, b) wavelet assessment [91]”.